# A Bayesian phase 2 model based adaptive design to optimise antivenom dosing: Application to a dose-finding trial for a novel Russell's viper antivenom in Myanmar

James A. Watson[1,2☯]*, Thomas Lamb[2,3☯], Jane Holmes[4], David A. Warrell[2], Khin Thida Thwin[5], Zaw Lynn Aung[5], Min Zaw Oo[6], Myat Thet Nwe[3], Frank Smithuis[2,3], Elizabeth A. Ashley[2,3,7]

1 Mahidol-Oxford Tropical Medicine Research Unit, Faculty of Tropical Medicine, Mahidol University, Bangkok, Thailand, 2 Centre for Tropical Medicine and Global Health, Nuffield Department of Medicine, University of Oxford, Oxford, United Kingdom, 3 Myanmar-Oxford Clinical Research Unit, Yangon, Myanmar, 4 Centre for Statistics in Medicine, Nuffield Department of Medicine, University of Oxford, Oxford, United Kingdom, 5 University of Medicine 1, Yangon, Myanmar, 6 University of Medicine 2, Yangon, Myanmar, 7 Lao-Oxford-Mahosot Hospital Wellcome Trust Research Unit, Vientiane, Laos

☯ These authors contributed equally to this work.
* jwatowatson@gmail.com

**Data Availability Statement:** The results in the manuscript are from simulated data. All underlying code can be found at: https://github.com/jwatowatson/AdaptiveAntivenomDesign.

## Abstract

For most antivenoms there is little information from clinical studies to infer the relationship between dose and efficacy or dose and toxicity. Antivenom dose-finding studies usually recruit too few patients (e.g. fewer than 20) relative to clinically significant event rates (e.g. 5%). Model based adaptive dose-finding studies make efficient use of accrued patient data by using information across dosing levels, and converge rapidly to the contextually defined 'optimal dose'. Adequate sample sizes for adaptive dose-finding trials can be determined by simulation. We propose a model based, Bayesian phase 2 type, adaptive clinical trial design for the characterisation of optimal initial antivenom doses in contexts where both efficacy and toxicity are measured as binary endpoints. This design is illustrated in the context of dose-finding for *Daboia siamensis* (Eastern Russell's viper) envenoming in Myanmar. The design formalises the optimal initial dose of antivenom as the dose closest to that giving a pre-specified desired efficacy, but resulting in less than a pre-specified maximum toxicity. For *Daboia siamensis* envenoming, efficacy is defined as the restoration of blood coagulability within six hours, and toxicity is defined as anaphylaxis. Comprehensive simulation studies compared the expected behaviour of the model based design to a simpler rule based design (a modified '3+3' design). The model based design can identify an optimal dose after fewer patients relative to the rule based design. Open source code for the simulations is made available in order to determine adequate sample sizes for future adaptive snakebite trials. Antivenom dose-finding trials would benefit from using standard model based adaptive designs. Dose-finding trials where rare events (e.g. 5% occurrence) are of clinical importance necessitate larger sample sizes than current practice. We will apply the model based design to determine a safe and

**Funding:** The MORU Tropical Health Network is funded by the Wellcome Trust. TL is on a fellowship funded by the Hamish Ogston Foundation. The funders had no role in study design, data collection and analysis, decision to publish, or preparation of the manuscript.

**Competing interests:** The authors have declared that no competing interests exist.

efficacious dose for a novel lyophilised antivenom to treat *Daboia siamensis* envenoming in Myanmar.

## Author summary

Snakebite envenoming is one of the most neglected tropical diseases considering its burden of mortality and morbidity. Antivenoms are the only known effective treatment for snake-bite envenoming but are frequently responsible for high rates of adverse reactions. Clinical development of antivenoms rarely follows the iterative phases of clinical development applied to other drugs. Dosing is typically based on pre-clinical testing. Here we propose a Bayesian model based adaptive design for phase 2 clinical trials aiming to determine the optimal dose of antivenom needed for treatment of snakebite envenoming. Optimality is defined using safety and efficacy thresholds contextual to the study. This design can be applied to all antivenoms which have binary efficacy and toxicity endpoints. Our design formally specifies a desired efficacy and a maximum tolerated toxicity. We use simulation studies to characterise adequate sample sizes to determine an approximately optimal dose under different scenarios. The simulation studies highlight the advantages of a model based design over simpler rule based alternatives. This design will be used to determine an effective and safe dose of the new lyophilised viper antivenom currently in use to treat Russell's viper envenoming in Myanmar.

## Introduction

Snake-bite envenoming (SBE) was re-categorized as a priority neglected tropical disease by the World Health Organization (WHO) in 2017 [1, 2]. Worldwide, there are as many as 2.7 million people affected by SBE resulting in an estimated 81,000 to 138,000 deaths per year [3–5], the burden of which disproportionately affects the poorest communities [5–7]. Antivenom is considered to be one of the most cost effective health interventions [8]. Despite this, due to challenges in manufacture, reliance on cold chain for transport and storage, and the geographically remote location of most envenomed patients, many patients do not receive the antivenom they require in a timely manner [9]. The 2019 WHO strategy for a globally coordinated response to SBE highlighted the need to prioritise clinical research into the safety and efficacy of antivenoms [2].

An integral part of the antivenom clinical research pipeline is pre-clinical assessment including the use of animal models. Pre-clinical assessment includes characterising the neutralisation of venom induced lethality and reversal of specific toxic effects of the venom, and antivenomics [10, 11]. Additional quantitative clinical assessment of antivenom pharmacokinetic properties (e.g. elimination half-life and volume of distribution) and pharmacodynamic properties (e.g. correction of coagulopathy, nephrotoxicity and haemodynamic instability) allows for the rational design of dosing strategies [12]. This is rarely done for antivenoms. Complementary to pharmacological consideration, dose optimisation can be done via phase 2 clinical trials. Ideally this is performed using adaptive design principles [13]. Adaptive designs are needed because it is rarely possible to pre-specify a suitably small set of doses that satisfy reasonable expectations for acceptable safety and efficacy. Many antivenoms will have narrow therapeutic windows and cannot be ethically administered to healthy volunteers, therefore dose optimisation trials need to simultaneously assess efficacy and toxicity.

Adaptive designs for dose-finding trials are of two main types. First, rule based designs which do not make any parametric assumptions regarding the relationship between the dose and the outcome of interest (e.g. efficacy or toxicity). A rule based design usually only assumes that there is a monotone increasing relationship between the dose and the outcome, i.e. the probability of the outcome increases with higher doses. The '3+3' design is the best known rule based adaptive design [14]. The standard formulation of the '3+3' design proceeds by recruiting successive cohorts of 3 subjects. Dose escalation for a subsequent cohort of 3 subjects occurs if no toxicity is observed amongst the previous 3; an additional 3 are given the same dose if toxicity is observed in only 1 out of 3; dose de-escalation occurs if toxicity is observed in 2 or more out of the previous 3. Rule based designs do not use information accrued across dosing levels, and therefore they have limited ability to rapidly identify the desired optimal dose with high confidence [15]. The alternative is a model based design, which requires determining a parametric relationship (model) between the dose and the outcome, termed a dose-response model [16]. The continual reassessment method [17] was the first proposed model based design for dose-finding. Data from sequentially enrolled patients are used to continually update the parameters of the dose-response model. Each enrolled patient is then assigned the expected optimal dose under the estimated dose-response model. The original rule based designs [14] and model based designs [17] for dose-finding were published around the same time. Although model based designs are more efficient, more flexible and have better operating characteristics [18], rule-based designs have been the dominant choice. For example, fewer than 1 in 10 trials in oncology—where dose-finding is critical—have used a model based approach [19, 20], mostly due to perceived difficulty of implementation and lack of understanding of the methods [21].

This paper outlines a model based, Bayesian adaptive design for phase 2 studies with the objective of optimising antivenom dosing. The structure of the design was motivated by the need to determine the optimal dosing for a novel antivenom developed to treat Russell's viper envenoming in Myanmar. Following a recent 4-year collaborative initiative between institutions in Myanmar and Australia entitled the *Myanmar Snakebite project*, antivenom production facilities improved, resulting in the production of a new monospecific lyophilised F(ab)2 antivenom (Viper antivenom BPI) [22]. This new lyophilised antivenom has replaced the former liquid antivenom and is now distributed countrywide. The lyophilised antivenom has the potential to greatly improve access to antivenom as the electrification rate in Myanmar is one of the lowest in Asia (70% in 2017 [23]). The current dosing strategy is based on unpublished results of pre-clinical testing and stratified into two doses according to absence or presence of clinical features of severity at presentation (80 mL and 160 mL, respectively). No clinical trial data or post marketing data have been published to support the efficacy or toxicity of these recommended doses.

This situation in Myanmar mirrors the development of many antivenoms worldwide [2, 24, 25] and highlights the need for high quality dose-optimisation studies. There is a need to standardise the methodology of clinical trials of antivenom whilst maintaining patient safety with robust patient monitoring built into study design. This calls for dose-finding phase 2 trials that can rapidly identify optimal dosing strategies, while minimising harm to patients. There are two concurrent considerations when optimising antivenom dosing. First, the efficacy of the dose, defined in the context of Russell's viper envenoming as the restoration of blood coagulation within 6 hours. Second, the dose-related toxicity, defined in this context as the occurrence of an anaphylactic reaction within 180 minutes post antivenom administration. Envenoming from different snakes will require different definitions of efficacy and toxicity. In the context of Russell's viper envenoming in Myanmar, the choice of these two binary clinical end-points was pragmatic due to their clinical significance, resource availability and replication of current

clinical practice. The model based design estimates dose-response curves for both the efficacy outcome and the toxicity outcome, and thus derives a contextually defined 'optimal dose'. The particularities of the design reported here were tailor-made for the dose-finding trial in *Daboia siamensis* envenoming but the design generalises to any systemic envenoming with clinically relevant endpoints whereby the efficacy and toxicity outcomes are both binary, e.g. [24, 26, 27]. We compared the *in silico* performance of this design against that of a tailor-made rule based design (modified '3+3' design or cumulative cohort design) under multiple simulation scenarios. The full simulation code written in R is open access and can easily be adapted to different antivenoms.

## Methods

### Model based adaptive design for dose-finding

In this section we give an overview of how doses are adaptively chosen during the trial and describe the necessary parameters for the adaptive assignment of doses to patients sequentially enrolled. First it is necessary to choose a randomisation ratio between the standard of care dosing arm and the adaptive dosing arm. This value can be set to 0 (i.e. all patients are assigned to the adaptive arm). Values greater than 0 result in a fixed proportion of patients assigned to the standard of care dose. This allows for a direct comparison (model free) between outcomes under the standard of care dose and outcomes under the dose to which the adaptive algorithm converges after a sufficient number of patients are enrolled. It also allows for a model free estimate of the efficacy and safety of the standard of care dose. The randomisation ratio is fixed throughout the trial.

In the adaptive arm, the adaptive dose assignment will depend on (i) the parametric dose-response models of toxicity and efficacy; (ii) the prior distribution over the model parameters; (iii) the toxicity and efficacy data observed for the antivenom in patients treated thus far; and (iv) the maximum tolerated toxicity and target efficacy (see below). The dose-efficacy and dose-toxicity models are updated using data from both the standard of care arm and the adaptive arm throughout the trial.

Patients are enrolled in successive cohorts of a pre-specified size $N_{\text{cohort}} \geq 1$. The choice of the value of $N_{\text{cohort}}$ is pragmatic as it determines how often it is necessary to update the model. Randomisation is performed at the individual level. We assume that the toxicity and efficacy outcomes for all previously enrolled cohorts of patients are known when a new cohort of patients is enrolled. At the start of the trial there is a "burn-in" period (a pre-specified number of patients). During this burn-in period, patients randomised to the adaptive arm are given the starting dose for the adaptive arm, which is the optimal dose under the prior distribution over the model parameters. After burn-in, patients randomised to the adaptive arm are given the posterior predicted optimal dose under the model (the distribution over model parameters is updated using all accrued data). If the model predicted dose is more than any previously trialled dose plus the maximum dose increment $\delta_\nu$, then patients are given the maximum allowed dose (the greatest previously assigned dose plus $\delta_\nu$).

For a given distribution over the model parameters, we define the optimal dose as follows. We first define a maximum tolerated toxicity (MTT), and a target efficacy level (TEL). The mean posterior predicted dose that has average toxicity equal to the MTT is denoted the maximum tolerated dose (MTD); and the mean posterior predicted dose that has average efficacy equal to the TEL is denoted the target efficacious dose (TED). The optimal dose is then defined as: $V^* = \min(\text{MTD}, \text{TED})$.

Additional parameters in the trial could include a minimum dose (the adaptively chosen dose cannot go below this dose); a maximum dose (the adaptively chosen dose cannot go

above this dose). If a minimum or a maximum dose are defined then these should be put into context with respect to the starting dose, the maximum dose increment or decrement and the total sample size. The purpose of a burn-in period for the adaptive arm is to reduce stochasticity at the start of the trial, especially in the context of weakly informative prior distributions over the model parameters. For example, a burn-in of 20 patients would imply that the adaptive arm would only be updated after the first 20 patients had been enrolled (irrespective of how they were randomised).

In addition, it is possible to specify stopping rules for the trial. For example, randomisation to the control arm (standard of care dose) could be stopped once sufficient evidence of its inferiority has been accrued (either too low and thus inferior efficacy, or too high and thus inferior due to excess toxicity) in comparison to the current adaptive dose. We would recommend the use of a non-parametric test (e.g. Fisher's exact test), with appropriate adjustment for multiple testing.

It may be the case that the antivenom used in the trial is from multiple batches. Batch variation can be an important contributor to variability in both toxicity and efficacy. It is straightforward to add a batch variation term in the adaptive models of efficacy and toxicity. This is an advantage of a model based design over a rule based design.

**Parametric models of toxicity and efficacy.**   The model based adaptive design necessitates determining parametric dose-response relationships for both the dose-related toxicity and the dose-related efficacy. We model both the efficacy and toxicity outcomes as Bernoulli random variables with dose-dependent parameters estimated under a generalised linear model. For the efficacy outcome we use the probit link function, and for the toxicity outcome we use the logistic link function.

The use of the probit link for the efficacy dose-response is motivated by a mechanistic understanding of how the antivenom acts. Assuming (i) there is a fixed linear relationship between the volume of venom in the body (which is unknown) and the dose of antivenom needed to neutralise all the circulating venom, and (ii) that the total mass of venom injected is approximately normally distributed, then the efficacy dose-response curve follows a normal cumulative distribution with mean $\mu$ and standard deviation $\sigma$ (i.e. probit link with parameters $\mu, \sigma$). The parameter $\mu$ corresponds to an efficacious dose of antivenom in 50% of patients. The value of $\mu + 1.64\sigma$ corresponds to an efficacious dose of antivenom in approximately 95% of patients. Weakly informative and interpretable priors can be set for both these parameters.

We choose to model the dose-toxicity relationship using a logistic function, where the dose is modelled on the logarithmic scale (base 2 for visual purposes, this does not impact the statistical inference). This is equivalent to fitting a Bayesian logistic regression model to the toxicity outcomes. Additional details of the Bayesian adaptive design are given in S1 Text.

## A modified '3+3' rule based design

In order to illustrate the advantages and disadvantages of model based adaptive designs, we compared the *in silico* performance of our model based adaptive design with that of a modified '3+3' rule based design. As in our model based design, patients are recruited in cohorts of size $N_{\text{cohort}}$. This is set to 3 in the classic '3+3' design, but in our case is a trial design parameter. The rule based design does not make parametric assumptions about the relationship between the dose and the outcomes. For each dose $v$ trialled, a dose-dependent frequentist estimate of toxicity, $\hat{\theta}_v^{\text{tox}}$, and a dose-dependent frequentist estimate of efficacy, $\hat{\theta}_v^{\text{eff}}$, are calculated. Based on these estimates, the dose is subsequently increased, decreased, or remains the same for the next $N_{\text{cohort}}$ patients, according to a pre-specified set of rules and trial design parameters (the MTT and the TEL). Our rule based design is a type of cumulative cohort design [28] as it uses

all the data from previous patients recruited to a particular dosing level. The traditional '3+3' design is memoryless (only uses information from the previous cohort). However, we refer to our rule based design as a modified '3+3' as this nomenclature is known more widely. In order for the two designs to be comparable, we use the same values for the MTT and TEL as in the model based design. We also use the same randomisation ratio between the adaptive arm and the standard of care arm. A detailed specification of the rules for adapting the dose is given in S2 Text.

A '3+3' type design has been used previously to identify candidate antivenom doses for treatment of envenoming by saw-scaled or carpet vipers (*Echis ocellatus*) [26]. However, the rules outlined in [26] were ambiguous. We also note that the design was for very small sample sizes (at most 6 per dosing level). We argue that small sample sizes do not allow for an accurate identification of the MTD or the TED when target event rates are 5% or 95%. Thus our proposed modification is a more appropriate comparator design for the performance of the model based design.

## A dose-finding trial in *Daboia siamensis* envenoming, Myanmar

The primary objective of our proposed study is to determine the optimal initial dose of the novel BPI lyophilised viper antivenom. The secondary objectives are to assess the relationship between the baseline venom concentration and the clinical outcomes, the sensitivity and specificity of the 20 minute whole blood clotting time (20WBCT) at detecting coagulopathy, the presence of ferryl-haem derivatives in the urine and the envenoming sequelae. Patients will be consented to participate if they present with a history of *Daboia siamensis* envenoming, are antivenom naive, aged ≥16 years and have a positive 20WBCT. Patients with a known coagulopathy will be excluded. All participants will have a serum venom assay to confirm envenoming performed retrospectively. Enrolled participants will be randomised to receive either standard of care (80 mL) or an adaptively chosen dose at a 1:4 ratio. The starting dose for the adaptive arm (120 mL) was determined using prior information for the new antivenom.

Aside from the initial dosing of antivenom, patients in each group will be managed according to Myanmar national guidelines [22]. Participants will be invited to attend a follow-up visit at 1 week and 3 months after discharge. This trial is registered at ClinicalTrials.gov with identifier number NCT04210141.

At the start of the trial, the optimal dose will be defined as the dose which either (i) restores blood coagulability at 6 hours in 95% of patients (efficacy endpoint), or (ii) causes anaphylaxis in 5% of patients (toxicity endpoint), whichever is lower. Both of these thresholds are subjective and were chosen after consultation with local clinicians and snakebite experts. The ambitious target of 95% efficacy was deemed appropriate for *Daboia siamensis* envenoming as delays in venom reversal increase the incidence of acute kidney injury [29]. In snake-bite envenoming involving species where the time of venom reversal is less critical, a lower efficacy threshold for initial dosing may be more appropriate. The reason for choosing an upper bound for the efficacy (here 95%) is that it is likely that the efficacy will plateau at higher doses. It is possible that an unknown proportion of patients will not meet the 6 hour efficacy endpoint as a result of delayed absorption of some venom components at the site of the bite [12]. Higher initial doses of antivenom may not solve this issue. These target efficacy and toxicity thresholds may not be the most appropriate for the context of Russell's viper envenoming in Myanmar. After recruitment of 50 and then 100 patients, we will perform full interim analyses and subsequently, in consultation with the Data and Safety Monitoring Board, update the target efficacy and target toxicity trial design parameters. For example, it may become apparent that 95% target efficacy is too high and that more than 5% of patients will not have restored

coagulation at 6 hours regardless of the dose. This interim analysis will allow for a re-adjustment of the trial parameters which decide the contextual target optimal dose.

The definition of efficacy for the purposes of this trial only pertains to the initial dose of antivenom. An efficacious dose is defined as restoring blood coagulability by 6 hours post administration as measured by the 20WBCT (a binary outcome). In case of treatment failure at 6 hours, all repeat doses will be equal to the initial dose administered. The total number of doses needed or the total time to restoration of coagulability will not be taken into account for the primary efficacy outcome for the adaptive design. The primary outcome used for determining toxicity of the initial dose is the occurrence of an anaphylactic reaction within 180 minutes of antivenom administration, as described by Sampson *et al* and accepted by the European Academy of Allergy and Clinical Immunology [30].

Only patients with 'non-severe' systemic envenoming will be invited to participate in the model based adaptive design study. Severely envenomed patients (who are estimated to represent 15% of envenomed patients) will be invited to participate in a parallel observational study. Participants and study doctors will not be blinded to treatment dose. All antivenom used in the study will be provided from a single batch. The antivenom will undergo retrospective pre-clinical testing to determine the median effective dose (ED50), the gold standard pre-clinical test for assessing antivenom efficacy [10, 11]. This will allow comparison between pre-clinical testing and *in vivo* outcomes.

The pragmatic addition of randomisation in this design is to ensure that the dosing range includes those recommended by current national guidelines which is the reference standard. This will allow for a model free comparison between the adaptively determined 'optimal' dose and the current national guidelines.

## Simulation study

We compared the stochastic behaviours of the model based and the rule based designs using a simulation study. Each simulation stopped after the enrolment of 260 patients. The global trial parameters were: $N_{cohort} = 3$; $\delta_v = 10$ mL; the starting dose in the adaptive arm was 120 mL; the standard of care dose was 80 mL; 20% of patients were randomised to the standard of care dose and 80% to the adaptive dose; the minimum assigned dose was set to 10 mL; no maximum dose was specified; the MTT was set to 5% and the TEL was set to 95%. No burn-in period was specified.

We simulated 2000 independent trials under seven scenarios, whereby each scenario specifies a simulation truth MTD and TED and underlying dose-toxicity and dose-efficacy relationships. In scenarios 1-4 the dose-response models for the model based design were well-specified. For scenarios 5-7 the dose-response models were mis-specified. Scenario 5 simulated data whereby toxicity was dose-independent. Scenarios 6-7 simulated data whereby the venom mass (and therefore the antivenom efficacy) was exponentially distributed. This implies that the distribution of venom mass has a heavier tail than predicted by the normal approximation. In all simulations all data, both from the adaptive and standard of care arms, were used in the model updates.

The seven scenarios are as follows:

1. The optimal dose is 'toxicity driven', and lower than our prior estimate (120 mL). By 'toxicity driven', we mean that the MTD is strictly less than the TED. In this scenario we choose an MTD of 80 mL, and a TED of 200 mL.

2. The optimal dose is 'efficacy driven', and lower than our prior estimate (120 mL). By 'efficacy driven', we mean that the TED is strictly less than the MTD. In this scenario, we chose a TED of 80 mL, and an MTD of 200 mL.

3. The optimal dose is 'toxicity driven', and higher than our prior estimate (120 mL). In this scenario, we choose an MTD of 300 mL, and a TED of 600 mL.

4. The optimal dose is 'efficacy driven', and higher than our prior estimate (120 mL). In this scenario, we chose a TED of 300 mL and an MTD of 600 mL.

5. Toxicity is idiosyncratic (dose-independent) and occurs in 15% of patients (mis-specified dose-toxicity relationship). The TED is 600 mL. In this scenario, as per our definition, the optimal dose is equal to the minimum dose (10 mL).

6. The optimal dose is 'efficacy driven' and lower than our prior estimate (120 mL). The efficacy dose-response curve is mis-specified (venom mass is exponentially distributed) and the TED is approximately 80 mL. The toxicity dose-response curve is well-specified with an MTD of 200 mL.

7. The optimal dose is 'efficacy driven' and higher than our prior estimate (120 mL). The efficacy dose-response curve is mis-specified (simulation truth is exponentially distributed) and the TED is approximately 300 mL. The toxicity dose-response curve is well-specified with an MTD of 600 mL.

All these simulations are fully reproducible via the code available on github: https://github.com/jwatowatson/AdaptiveAntivenomDesign. A static version of the code can be found at https://doi.org/10.5281/zenodo.3931615. The exact parameters for each of the simulation scenarios are given in the code. The code is provided in a modular format to ease adaptation to different contexts.

## Literature review of antivenom dose-finding trials

A systematic review of the literature relating to dose-finding trials of antivenom was performed using the Medline medical database. The following keywords "antiven*" and "dose-finding" or "clinical trial" were used and searched for on the first of November 2019. Only papers reporting a comparative trial of two or more doses of the same antivenom were included. Papers reporting pre-clinical trials and studies comparing two or more different antivenoms were excluded. Referenced articles in the articles identified by the search were assessed for suitability and included if they met the outlined criteria. The final set of studies reviewed are shown in S1 Table.

## Results

### Literature review on antivenom dose-finding trials

Using the search criteria, 112 abstracts were reviewed for suitability. Sixteen papers were identified as including clinical data relating to two or more doses of antivenom. Studies included a combined total of 1165 envenomed patients from a variety of taxonomic orders (Scorpiones (1), Lepidoptera (1), Hymenoptera (1) and Squamata sub order Serpentes (13)), see S1 Table. One paper described a trial protocol for which the results have yet to be published. Five papers investigated new antivenoms with the remaining nine papers investigating established antivenoms. Four papers performed a retrospective review of antivenom doses and 10 studies were conducted/plan to be conducted prospectively. Relating to dose-finding trial design, one paper used a '3+3' dose escalation model [26] while the remaining 13 papers all assessed pre-determined doses. None of the identified studies used a model based adaptive design. Seven of the dose-finding studies referenced preclinical data to assist with initial antivenom dosing.

There was a lack of consistency of clinical endpoints. Clearly defined efficacy and toxicity endpoint outcomes were described in one randomised clinical trial of neurotoxic snake-bite. In 13 papers with predominant haemotoxic venom, the clinical endpoints used were clotting time (5), 20WBCT (5) prothrombin time (1), fibrinogen (1) and unspecified complications (1). Nine papers did not specify toxicity as an endpoint whilst six studies used undefined adverse reactions as an endpoint. Two papers conducted power calculations and a further three papers conducted phase 1/2 trials with small sample sizes using new candidate antivenoms. The remaining eleven papers did not include power calculations or estimations of sample size.

## Simulation study comparing rule based and model based designs

To understand the advantages and disadvantages of the model based design in comparison to the simpler rule based design, we ran seven independent simulation studies, each simulating 2000 independent trials with 260 patients. All the simulation studies were written in view of the Myanmar dose-finding trial for Russell's viper envenoming, whereby 260 is considered a maximum feasible sample size. The source code used in these simulation studies is openly accessible and easily modifiable for application to different contexts. In all the simulation scenarios considered, the model based design converged to the simulation optimal dose faster than the rule based design (Fig 1), including when the models used by the model based design were mis-specified (Fig 1, panels 5-7).

Simulation scenario 4 is the most likely scenario in the Myanmar context. In this scenario, the antivenom has an MTD higher than its TED. Fig 2 shows the *in silico* stochastic behaviours of the model based design and the rule based design under scenario 4. Under the model based design, the mean trial assigns approximately the correct dose after enrolling 100 patients. Correct estimation of the MTD necessitates more than double the number of patients given it is a very rare event at the trialled doses (approx. 1% occurrence). In comparison, under the rule based design, the average trial is still assigning lower than optimal doses after 260 patients are enrolled. At the 260th patient, 91% of the simulated trials under the model based design assign a dose within ±10% of the optimal dose. Only 62% of the rule based designs assign a dose within ±10% of the optimal dose.

Fig 3 shows the behaviour of the designs for scenario 1, whereby the starting dose is in fact higher than the optimal dose, determined in this scenario by the MTD. This scenario corresponds to the situation whereby the manufacturing of the antivenom is not as good as reported (the reported low toxicity of the novel BPI antivenom in Myanmar is from anecdotal evidence only) and causes frequent (approximately 10% occurrence) toxicity at the starting dose in the adaptive arm. Convergence to approximately the optimal dose occurs for the average trial under the model based design after 100 patients are recruited. In comparison, the rule based design converges is approximately three times slower to converge to the optimal dose.

Understanding how model mis-specification impacts the convergence of the model based design is important for implementation. We simulated the stochastic behaviour of the trial designs when (i) toxicity is not dose dependent (as argued by [31]) and (ii) when the distribution of venom mass is considerably different from a normal approximation. In both of these settings, the model based design outperforms the rule based designs. In scenario 5, anaphylaxis occurred in 15% of patients independently of the dose administered. This is an extreme scenario but representative of a poorly manufactured antivenom and serves to highlight how the designs adapt to this lack of an underlying dose-response, which is a key assumption for both the rule based design and the model based design. Under our definition

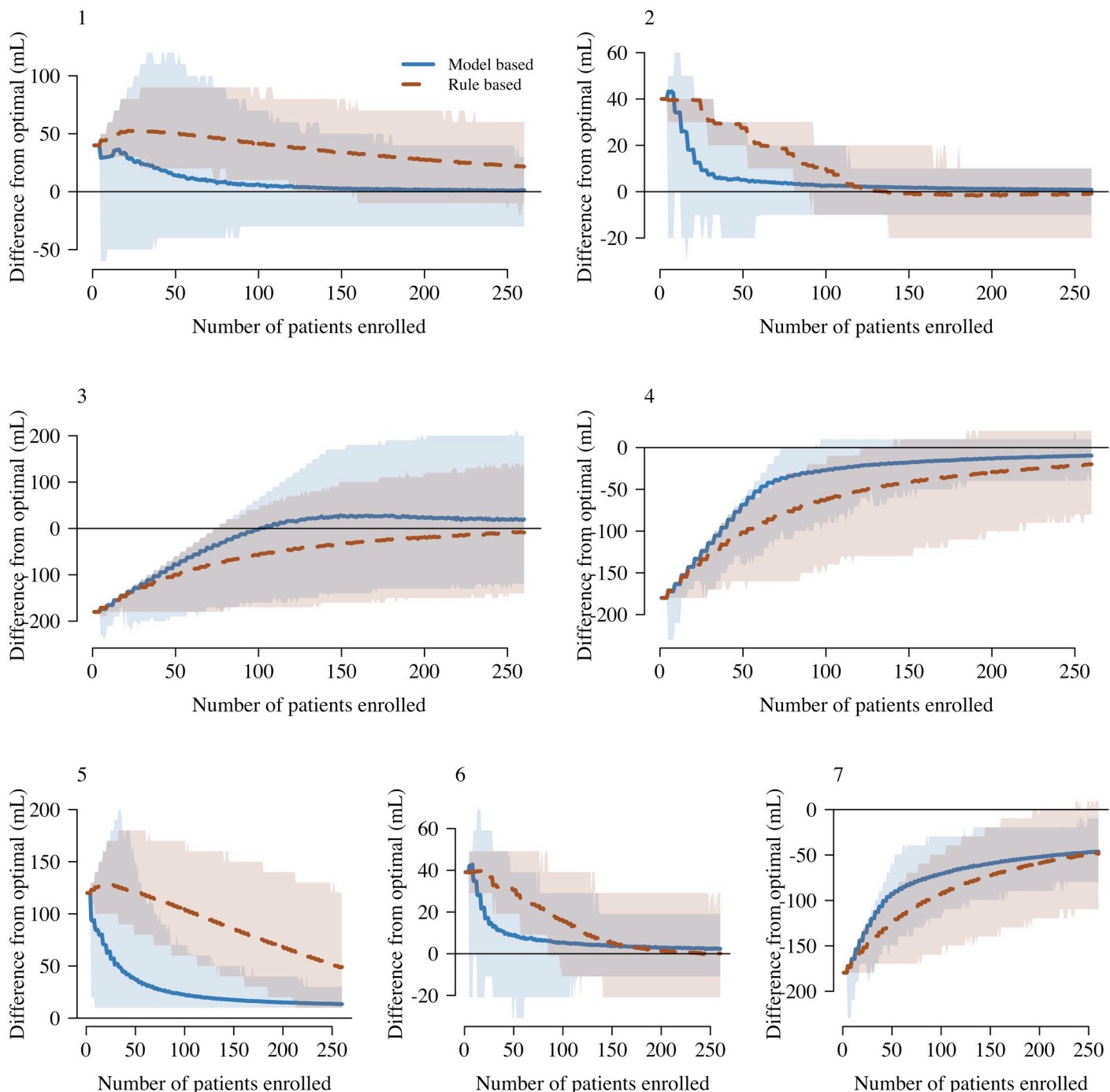

**Fig 1. Comparison between the model based design (blue) and the rule based design (red) across all simulation scenarios.** Panel numbers correspond to the simulation scenario defined in the Methods section. In each panel, the thick lines (shaded areas) show the mean difference (95% interval of variation across trials) between the assigned doses and the simulation true optimal dose. Panels 1-4 show the results for the well-specified scenarios; panels 5-7 for the mis-specified scenarios. Note that each panel has a different y-axis range and the horizontal line shows the 0 y-axis value for reference.

of 'optimality', in this scenario, any non-zero dose is above the MTD and therefore the optimal dose is 0 mL. Both the model based design and the rule based design converge towards the minimum allowed dose (10 mL), but the convergence is much faster for the model based design (Fig 1, panel 5).

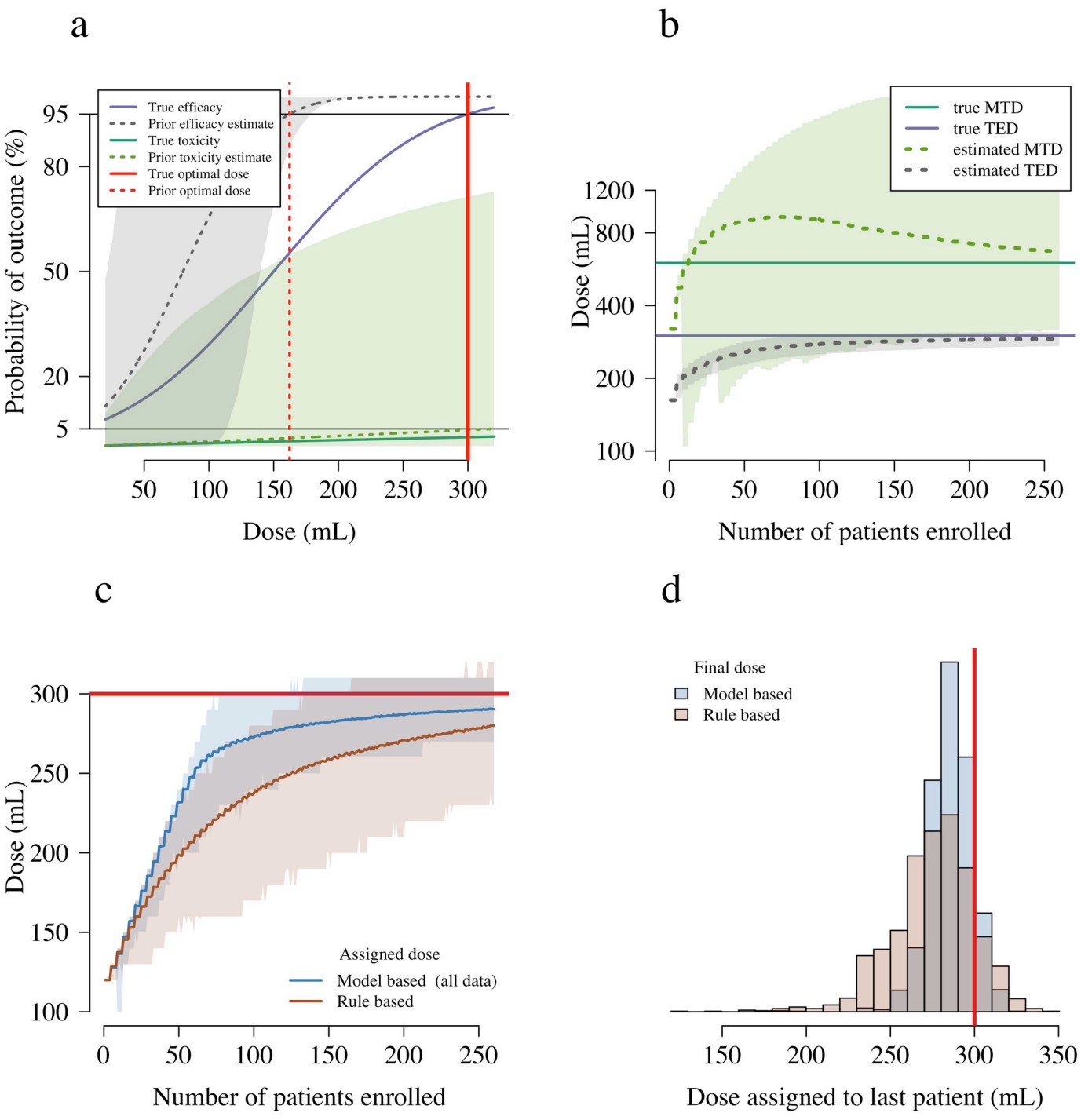

**Fig 2. Operating characteristics of the rule based and model based designs under the simulation scenario 4.** In this scenario the simulation optimal dose is the TED (300 mL), shown as a thick red line in panels a, c and d. Panel a shows the simulation truth (thick lines) and the prior distributions used in the model based design (dashed lines: mean prior estimate; shaded areas: 90% credible interval). Panel b shows the evolution of the estimated MTD and TED as a function of the number of patients enrolled (dashed lines: estimate in the average trial; shaded areas: 90% interval of variation across trials). Panel c shows the assigned doses for each design: the thick lines show the assigned dose in the average trial; the shaded areas show 90% intervals of variation across trials. Panel d compares the distributions of the final assigned doses for the two designs.

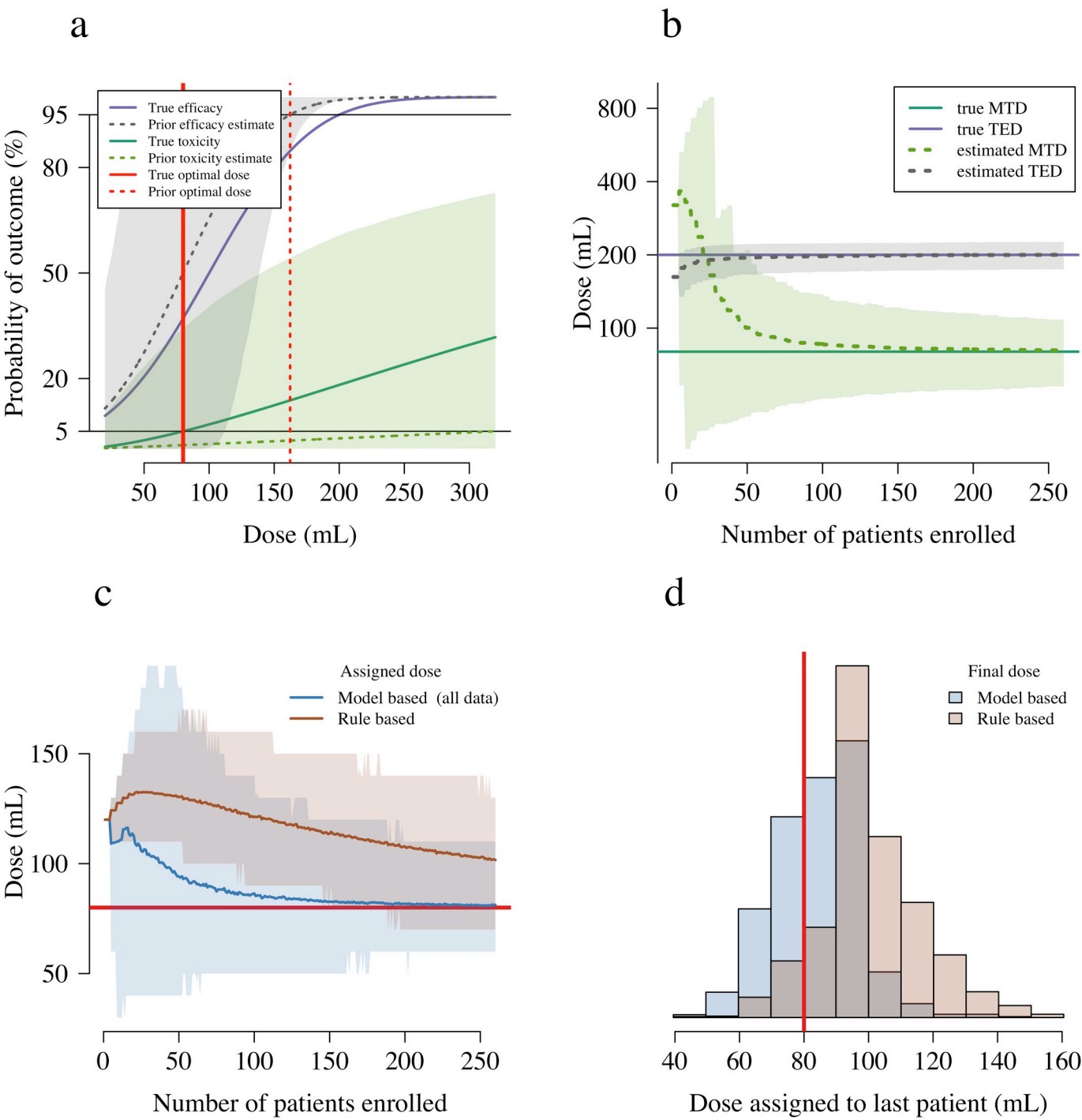

**Fig 3. Operating characteristics of the rule based and model based designs under the simulation scenario 1.** In this scenario the simulation optimal dose is the MTD (80 mL), shown as a thick red line in panels a, c and d. Panel a shows the simulation truth (thick lines) and the prior distributions used in the model based design (dashed lines: mean prior estimate; shaded areas: 90% credible interval). Panel b shows the evolution of the estimated MTD and TED as a function of the number of patients enrolled (dashed lines: estimate in the average trial; shaded areas: 90% interval of variation across trials). Panel c shows the assigned doses for each design: the thick lines show the assigned dose in the average trial; the shaded areas show 90% intervals of variation across trials. Panel d compares the distributions of the final assigned doses for the two designs.

## Discussion

Antivenoms for the management of SBE have been subjected to few of the sequential clinical studies required of new therapeutics for other diseases, both prior to and post licensing. As a result, most antivenom dosing is based on animal models that are known to extrapolate poorly to humans [10]. As for the development of many novel oncology therapies, due to the high risk of adverse reactions related to antivenom and lack of alternative therapeutic options, conventional phase 1 clinical trials in healthy volunteers to establish safety are considered unethical [32]. The unknown rate of toxicity and possibly narrow therapeutic window further underlines the need for well designed dose-finding trials with well defined efficacy and toxicity endpoints. This position is supported by the WHO who have highlighted the need to prioritise clinical research into the safety and effectiveness of antivenoms [2].

Model based adaptive designs have clear benefits over the simpler rule based designs such as the '3+3' design. The superior operating characteristics and greater efficiency are well described in the literature [18]. Our simulations support this and we provide a flexible model framework for antivenom dose-finding trials that wish to simultaneously optimise efficacy whilst guaranteeing safety. By pre-specifying a MTT patient safety is inherently built in to the study design. We note that concurrent consideration of both toxicity and efficacy in phase 1-2 trials is not novel in itself. Although most adaptive dose-finding designs consider a single outcome variable (either toxicity or efficacy), there are important research areas such as oncology, where concurrent consideration of both outcomes is more efficient and possibly safer [33]. Explicitly modelling both the toxicity and efficacy dose-response curves has been previously proposed, in both a frequentist setting [34] and a Bayesian setting [35, 36]. The main contribution of this work is to (i) highlight the utility of model based adaptive designs in the specific context of antivenom dose-finding, (ii) propose dose-response models for both toxicity and efficacy which are appropriate and meaningful in the context of antivenom dose-finding, and (iii) provide open source software that can be re-used for the design of other trials.

### The importance of choosing the correct efficacy and toxicity endpoints

The model based adaptive design described and illustrated in this paper has the potential for widespread uptake in the design of trials of antivenom to treat bites of other snake and venomous species resulting in coagulopathy, neurotoxicity and local tissue effects. The design is dependent on consistent and clinically relevant efficacy and toxicity endpoints. Clinically relevant efficacy endpoints for envenoming resulting in coagulopathy include the 20WBCT (applicable to Myanmar where other clotting assays are not available) and a clinically significant elevation in prothrombin time or INR. The objective clinical endpoint used by Alirol *et al.* in neurotoxic envenoming could be adapted as an efficacy endpoint in an adaptive design, i.e. improvement in FEV1 or the avoidance of ventilation from a starting position of pre-defined neurological impairment [27]. These simulation studies for both the model based and rule based designs highlight the need for larger phase 2 type dose-finding trials. Dose-finding trials have traditionally been viewed as small clinical studies recruiting 10-20 patients. However, if rare events (approximately 5% occurrence or less) are of key clinical importance, then sample sizes in the low hundreds are needed. In our setting we believe that a sample size of approximately 200 patients would be sufficient to accurately determine the optimal dose of the new Russell's viper antivenom. This contrasts to a median of 50 patients in the surveyed literature (S1 Table).

The mechanism of early anaphylactic reactions secondary to antivenom administration, and the relationship to dose is poorly understood [37]. The lack of prior exposure to antivenom in the majority of patients with anaphylactic reactions to antivenom and poor

predictive value of skin testing is suggestive of alternative mechanisms to classical Type 1 IgE mediated hypersensitivity [38]. Stone *et al.* examined cytokine activity, complement activity and mast cell degranulation in patients envenomed by a *Daboia russelii* before and after antivenom therapy [39]. The study demonstrated elevated levels of complement activity and inflammatory mediators before antivenom and subsequent rises in mast cell tryptase and histamine following antivenom in keeping with mast cell degranulation. As described earlier, clinical trials in antivenom therapy rarely define clear toxicity endpoints and are not powered to accurately characterise rare events (i.e. those occurring in 5% or fewer patients). In the dose-finding trials displayed in S1 Table, six studies demonstrate a trend suggestive of a dose-toxicity relationship [26, 27, 40–43], two studies show the opposite relationship [44, 45] and the remaining eight studies do not report toxicity as an outcome. An earlier study by Reid with large numbers of patients was suggestive of a dose-toxicity relationship [46]. Our design hypothesises that toxicity is dose dependent following a logistic curve as a function of the logarithm of the dose. We show that our design is robust to mis-specification in this model, both for the model of efficacy (not logistic) and for the toxicity model (when the anaphylactic reactions occur idiosyncratically with no dose toxicity relationship). However, care is needed in determining what is an acceptable rate of toxicity in the trial (the MTT).

## Limitations of the Bayesian model based adaptive design

Our proposed adaptive design has some important limitations. First and foremost it is necessary to pre-specify a TED and a MTT. Lack of prior knowledge of how the antivenom acts may result in setting unrealistic values for the TED and the MTT. We note that this affects the Bayesian model based design and the rule based design equally. To alleviate this concern we propose interim analyses (after 50 and then 100 patients are enrolled) with the specific purpose of re-evaluating the TED and the MTT for futility. An alternative to setting hard thresholds for the TED and MTT is to use an EffTox dose-finding design [47, 48]. EffTox necessitates the construction of utility contours which determine an explicit trade-off between efficacy and toxicity. We chose not to use this design for this context as it is was not possible to reach an agreement with clinical experts and because it made a strict comparison with a rule based method impossible. However, the EffTox design may be appropriate for other snakebite dose-finding trials. A limitation specific to the Bayesian design is the need to specify prior distributions over the dose-efficacy and dose-toxicity models. Prior elicitation is difficult, especially with limited numbers of published clinical trials, lack of post marketing data and antivenom batch variability. It is important to note that the priors do not need to be strongly informative but serve to minimise stochasticity at the start of the trial. Specifying an adequate "burn-in" period alleviates the problem of poorly specified priors.

A well known disadvantage of a model based design is the need for greater statistical support in the study design and continuous support during subject enrolment to determine each sequential adapted dose. The simulation scripts that we provide should serve to help trial statisticians design Bayesian model based adaptive trials in different contexts. In addition, the use of real-time electronic case reporting allows for remote, off-site statistical support.

Despite these limitations, we believe that this Bayesian model based design is particularly pertinent to assessing the optimal dose of BPI Viper Antivenom for *Daboia siamensis* envenoming in Myanmar. *Daboia siamensis* envenoming remains a significant health burden in Myanmar resulting in considerable morbidity and mortality [29, 49]. Fast identification of an efficacious and safe dose means that few patients in the trial will be administered sub-optimal doses. Indeed, using the incrementally accrued data during the study should result in a comparatively low sample size. This is an important consideration given the rural nature of

envenoming and cost of implementing studies at multiple sites. The resultant description of dose-efficacy and dose-toxicity relationships will enable policy makers to confidently choose a dose which provided satisfactory efficacy and is safe in the majority of patients.

## Future work

The design of a dose-finding study is only as good as the endpoint which determines the treatment response. For example, using the 20WBCT as a surrogate marker of envenoming at presentation, and then resolution of envenoming at 6 hours, introduces variability and potential bias for which the design cannot account. Improving dose-finding studies for novel antivenoms will rely critically on improving pharmacodynamic endpoints and understanding their relationship to antivenom pharmacokinetics. We advocate the use of model based adaptive designs for dose-finding but this solves only part of the problem. There is serious need for future work that develops more specific pharmacodynamic markers of treatment efficacy in SBE from Russell's viper and haemotoxic snakes more widely.

## Supporting information

**S1 Table. List of studies selected from literature review.**
(PDF)

**S1 Text. Bayesian adaptive design details.**
(PDF)

**S2 Text. '3+3' (cumulative cohort) adaptive design rules.**
(PDF)

## Author Contributions

**Conceptualization:** James A. Watson, Thomas Lamb, Jane Holmes, David A. Warrell, Frank Smithuis, Elizabeth A. Ashley.

**Formal analysis:** James A. Watson.

**Funding acquisition:** Thomas Lamb, David A. Warrell, Frank Smithuis.

**Investigation:** James A. Watson.

**Methodology:** James A. Watson, Thomas Lamb, Jane Holmes, Frank Smithuis, Elizabeth A. Ashley.

**Project administration:** James A. Watson, Elizabeth A. Ashley.

**Resources:** James A. Watson.

**Software:** James A. Watson.

**Supervision:** Jane Holmes, David A. Warrell, Elizabeth A. Ashley.

**Validation:** Khin Thida Thwin, Zaw Lynn Aung, Myat Thet Nwe, Frank Smithuis, Elizabeth A. Ashley.

**Writing – original draft:** James A. Watson.

**Writing – review & editing:** James A. Watson, Thomas Lamb, Jane Holmes, David A. Warrell, Khin Thida Thwin, Zaw Lynn Aung, Min Zaw Oo, Myat Thet Nwe, Frank Smithuis, Elizabeth A. Ashley.

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
