## [Decision Letter · Decision Letter 0]

20 May 2020

Dear Dr Watson,

Thank you very much for submitting your manuscript "A Bayesian phase 2 model based adaptive design to optimise antivenom dosing: application to a dose-finding trial for a novel Russell's viper antivenom in Myanmar" for consideration at PLOS Neglected Tropical Diseases. As with all papers reviewed by the journal, your manuscript was reviewed by members of the editorial board and by several independent reviewers. In light of the reviews (below this email), we would like to invite the resubmission of a significantly-revised version that takes into account the reviewers' comments. 

It was difficult to collect reviews amid the pandemic, but at last we succeeded in obtaining four expert reviews that, in our opinion, adequately cover the interdisciplinary breadth of the work presented. 

As all reviewers agree regarding the potential importance of this work beyond the Myanmar case study with Daboia siamensis/antivenom, we request a major revision that appropriately addresses all suggestions and queries by these reviewers, and adequately honours any comparable or similar published approaches. This should also include an expanded, full and transparent discussion of the (actual, perceived, potential) limitations of this case study. Inclusion criteria and choice of endpoints are issues that were mentioned by reviewers, and these should consequently be addressed in greater detail. More than one reviewer got stuck at the 20WBCT "controversy" (if there is any) and restoration of blood coagulability after 6 hr (which might not be a predictor of either survival or no need for dialysis or amputation, depending on the particular antivenom used, and evidence to be shown), so more room should be allocated to explaining and discussing this.

We cannot make any decision about publication until we have seen the revised manuscript and your response to the reviewers' comments. Your revised manuscript is also likely to be sent to reviewers for further evaluation.

Sincerely,

Jean-Philippe Chippaux, M.D., Ph.D.

Deputy Editor

Jean-Philippe Chippaux

Deputy Editor

It was difficult to collect reviews amid the pandemic, but at last we succeeded in obtaining four expert reviews that, in our opinion, adequately cover the interdisciplinary breadth of the work presented. 

As all reviewers agree regarding the potential importance of this work beyond the Myanmar case study with Daboia siamensis/antivenom, we request a major revision that appropriately addresses all suggestions and queries by these reviewers, and adequately honours any comparable or similar published approaches. This should also include an expanded, full and transparent discussion of the (actual, perceived, potential) limitations of this case study. Inclusion criteria and choice of endpoints are issues that were mentioned by reviewers, and these should consequently be addressed in greater detail. More than one reviewer got stuck at the 20WBCT "controversy" (if there is any) and restoration of blood coagulability after 6 hr (which might not be a predictor of either survival or no need for dialysis or amputation, depending on the particular antivenom used, and evidence to be shown), so more room should be allocated to explaining and discussing this.

Reviewer's Responses to Questions

**Key Review Criteria Required for Acceptance?**

**Methods**

-Are the objectives of the study clearly articulated with a clear testable hypothesis stated?

-Is the study design appropriate to address the stated objectives?

-Is the population clearly described and appropriate for the hypothesis being tested?

-Is the sample size sufficient to ensure adequate power to address the hypothesis being tested?

-Were correct statistical analysis used to support conclusions?

-Are there concerns about ethical or regulatory requirements being met?

Reviewer #1: This is a well written research article. Even though I am not an expert in modelling, I can understand the logic of developing the model.Objectives are clearly presented. The study design is appropriate and the statistical methods are appropriate. There are no ethical concerns in this paper.

Reviewer #2: The authors developed a Bayesian phase 2 model for characterizing the optimal initial antivenom doses, considering toxicity and efficacy endpoints. The rational of the designed model is understandable. Exclusive and inclusive criteria are considererd, despite the chosen resolution of envenoming at 6 hours as endpoint ids not sufficient.

The logistic function parameterization of the dose-response model and scenarii are interesting.

Reviewer #3: The objectives of the study are clearly articulated with a clear testable hypothesis stated, the study design is appropriate to address the stated objectives, the population are clearly described and appropriate for the hypothesis being tested and the sample size is sufficient to ensure adequate power to address the hypothesis being tested. Statistical analysis are correct, ethical and regulatory requirements are met

Reviewer #4: The description of the trial design needs more details. the models for efficacy and toxicity are both just GLMs with a probit and logit link respectively. But how are these used? That's not at all clear. presumably several doses are used at the start, but then how are new doses chosen? We need to see more details.

**Results**

-Does the analysis presented match the analysis plan?

-Are the results clearly and completely presented?

-Are the figures (Tables, Images) of sufficient quality for clarity?

Reviewer #1: yes

It may be useful for the common reader if the presentation of results can be simplified to make better understanding of the results.

Figures are clear and of very good quality.

Reviewer #2: Results are described in detail and correlations are depicted well.

In the discussion part, some additional details are given to the assessed data.

Reviewer #3: The presented analysis match the analysis plan and the results are clearly and completely presented. Also the figures and their quality are sufficiently clear

Reviewer #4: The presentation of the results is OK, but I wonder if it could be done more succinctly, by defining a (possibly artificial, i.e. purely for the presentation of the results) stopping rule and presenting how long it takes the methods to reach the rule under the different scenarios. This could then be presented in a single plot, which would be easier to interpret.

**Conclusions**

-Are the conclusions supported by the data presented?

-Are the limitations of analysis clearly described?

-Do the authors discuss how these data can be helpful to advance our understanding of the topic under study?

-Is public health relevance addressed?

Reviewer #1: The limitation of the study is clearly the use of the WBCT20 as a surrogate. The authors have addressed this as a limitation and is acceptable.

Reviewer #2: The limitation of the analysis to the chosen resolution of envenoming at 6 hours is reported and future development is addressed.

Dose-finding trials would benefit from the developed model based on adaptative designs.

Reviewer #3: The authors show clearly the relevance of their results especially on the management of snake envenomation which is a public health problem in many countries. However, in this paper, authors did not well discuss the limitations to the approach. The conclusion lacks precisions.

Reviewer #4: I think the onclusions are OK, although is seems odd to end the discussion with a coments about the 20WBCT criterion. That's not what this manuscript is about, so I don't see that it's necessary.

**Editorial and Data Presentation Modifications?**

Reviewer #1: Since the readership of this article is highly likely to be clinicians, it would be appropriate to simplify and shorten the methods section if possible.

Reviewer #2: The manuscript submitted by Watson et al., entitled "A Bayesian phase 2 model based adaptive design to optimise

antivenom dosing: application to a dose-finding trial for a novel Russell's viper antivenom in Myanmar" is written in basic language and is well understandable. 

The study is quite relevance: Antivenom's administration is a critical step in healthcare management of SBE, one of the important NTDs. For safety administration, small and effective dose is required and has to be carefully assessed. Simulation approach is useful to better control adverse effects. 

The study is well performed and assessed data covers a broad spectrum and includes necessary controls. 

Minor comment:

- In the Introduction section it will be useful to describe all the pharmacodynamic markers that are important to be considered, independently of the conducted study.

- It remains unclear whether the introduced model adapted to dosing Russell's viper envenoming could be standardized/globalized to other lyophilized or liquid SBE specific Antivenoms, taking into consideration the process of antivenom's preparation, manufacturing, and batch to batch differences.

Reviewer #3: In this paper, authors did not well discuss the limitations to the approach. Indeed, all Bayesian analyses are dependent on the choice of prior distributions, which can be erroneous. In this study, the prior distributions for the dose-efficacy and dose-toxicity response models and the longitudinal model were kept weak so that the posterior distributions would be mostly shaped by the emerging data. In addition, authors could not be certain that each of the seven simulated scenarios was equally likely, so it is possible that their estimation for the average expected sample size, which was not weighted, might not be accurate

Minor Comments

1/ One section combing Author Summary and Abstract sections is preferable. 

2/ Authors defined efficacy as the restoration of blood coagulability within six hours, and toxicity as anaphylaxis For Russell’s viper. Can they explain on which basis they chose these parameters. 

3/ Line 382 : As a result, most antivenom dosing is based on animal models that are known to extrapolate poorly to humans. Authors should highlight also the importance of animal experimentation as preclinical phase

4/ Why authors did not discuss the possible scenario in the discussion part ; this weakness is the lack of a deeper discussion. 

5/ References should be updated exp : Line 3 page 2 : « Worldwide, there are as many 3 as 2.7 million people affected by SBE resulting in an estimated 81,000 to 138,000 deaths 4 per year »

Reviewer #4: I would suggest the authors re-arrange the manuscript to emphasise the generality of their work. After the introduction, they should present their adaptive ddesign is a generic way, so that it can be accessible to a wider group (basically, anyone wanting to efficiently find an optimum dose when they have to worry about toxicity). After that, present the D. siamensis envenoming context as a justification for the simulation study.

I was slightly surprised that this problem hadn't been looked at before, and a bit of searching suggests that there has been work in this area (e.g. DOI: 0.1002/sim.6124, 10.1080/10543400903280613). The seminar work appears to be DOI: 10.2307/2533268, but it doesn't consider a dose response. I think this body of work need sto be mentioned (I suspect there is more work than this - some might have been done with phase I trials).

Minor Comments

l80: Is that the first of November, or the 11th of January?

l198-208: This is then the same as a GLM with a binomial response and a probit link. 

l207: \\mu + 1.64\\sigma for 95% efficacy, I think.

l216-8: What about the slope?

**Summary and General Comments**

Reviewer #1: Antivenom dosing studies are rare and when new antivenoms are produced, establishing dosing is always challenging. This article provides new insight into this area from which others can learn.

Reviewer #2: The manuscript submitted by Watson et al., entitled "A Bayesian phase 2 model based adaptive design to optimise

antivenom dosing: application to a dose-finding trial for a novel Russell's viper antivenom in Myanmar" is written in basic language and is well understandable. 

The study is quite relevance: Antivenom's administration is a critical step in healthcare management of SBE, one of the important NTDs. For safety administration, small and effective dose is required and has to be carefully assessed. Simulation approach is useful to better control adverse effects. 

The study is well performed and assessed data covers a broad spectrum and includes necessary controls. 

Minor comment:

- In the Introduction section it will be useful to describe all the pharmacodynamic markers that are important to be considered, independently of the conducted study.

- It remains unclear whether the introduced model adapted to dosing Russell's viper envenoming could be standardized/globalized to other lyophilized or liquid SBE specific Antivenoms, taking into consideration the process of antivenom's preparation, manufacturing, and batch to batch differences.

Reviewer #3: The article entitled « A Bayesian phase 2 model based adaptive design to optimise antivenom dosing: application to a dose-finding trial for a novel Russell's viper antivenom in Myanmar » proposes a Bayesian model based adaptive design for clinical trials aiming to determine the optimal dose of BPI antivenom. The main motivation behind this work is the bias of some clinical studies induced by small sample sizes or ethical constraints as well as the non-standardization of the dose selection for treating envenomation to maintain both efficacy and safety of the treatment. The authors use the case of Daboia siamensis (Eastern Russell’s viper) as an illustration for the application of their method. They argued the effectiveness of "Bayesian adaptive design" over the simpler rule-based design. As far as I know, this is the first application of the Bayesian adaptive design in antivenom research. Thanks to this model, it is possible to integrate prior information from previous clinical studies. The authors did this by mining the literature which identifies 16 papers by querying the Medline database followed by filtering. Therefore, the paper is original and brings a new perspective to a topic that represents a challenge in many countries. From the results, it appears that the model-based design performs better than the simpler ‘3+3’ design. The simulation made in this work showed a quicker convergence of the model-based design except for one (scenario 7) were no model was able to converge. The results showed also a contradiction to previous works about the effective size of the cohort to accurately determine the optimal dose which adds to the value of the author's work. However the conclusion as well as the discussion of the results are not so clear.

In conclusion, overall the paper reads well, the methods are described clearly and the results are well presented. For that this work deserves consideration, however, the current version of the manuscript is affected by some limits.

Reviewer #4: I must apologise for the delay in providing my comments, the current situation has hardly been ideal I'm afraid.

I think this is OK, but the method needs to be presented more clearly. It would be impossible to replicate it just from the description.

PLOS authors have the option to publish the peer review history of their article (what does this mean?). If published, this will include your full peer review and any attached files.

Reviewer #1: Yes: Indika Gawarammana

Reviewer #2: Yes: Balkiss Bouhaouala-Zahar

Reviewer #3: No

Reviewer #4: No
---

## [Decision Letter · Decision Letter 1]

10 Oct 2020

Dear Dr Watson,

We are pleased to inform you that your manuscript 'A Bayesian phase 2 model based adaptive design to optimise antivenom dosing: application to a dose-finding trial for a novel Russell's viper antivenom in Myanmar' has been provisionally accepted for publication in PLOS Neglected Tropical Diseases.

Best regards,

Jean-Philippe Chippaux, M.D., Ph.D.

Deputy Editor

Jean-Philippe Chippaux

Deputy Editor

Reviewer's Responses to Questions

**Key Review Criteria Required for Acceptance?**

**Methods**

-Are the objectives of the study clearly articulated with a clear testable hypothesis stated?

-Is the study design appropriate to address the stated objectives?

-Is the population clearly described and appropriate for the hypothesis being tested?

-Is the sample size sufficient to ensure adequate power to address the hypothesis being tested?

-Were correct statistical analysis used to support conclusions?

-Are there concerns about ethical or regulatory requirements being met?

Reviewer #4: NA

**Results**

-Does the analysis presented match the analysis plan?

-Are the results clearly and completely presented?

-Are the figures (Tables, Images) of sufficient quality for clarity?

Reviewer #4: NA

**Conclusions**

-Are the conclusions supported by the data presented?

-Are the limitations of analysis clearly described?

-Do the authors discuss how these data can be helpful to advance our understanding of the topic under study?

-Is public health relevance addressed?

Reviewer #4: NA

**Editorial and Data Presentation Modifications?**

Reviewer #4: NA

**Summary and General Comments**

Reviewer #4: My apologies for not being able to review this fully. From a (quick) read of the manuscript and the response to the previous round of reviews, I think this is fine. I hope this work will be useful in practice for a variety of SBEs.

PLOS authors have the option to publish the peer review history of their article (what does this mean?). If published, this will include your full peer review and any attached files.

Reviewer #4: No

---

## [Editor Report · Acceptance letter]

10 Nov 2020

Dear Dr Watson,

We are delighted to inform you that your manuscript, "A Bayesian phase 2 model based adaptive design to optimise antivenom dosing: application to a dose-finding trial for a novel Russell's viper antivenom in Myanmar," has been formally accepted for publication in PLOS Neglected Tropical Diseases.

Best regards,

Shaden Kamhawi

co-Editor-in-Chief

Paul Brindley

co-Editor-in-Chief
